# Position: Maximizing Neural Regression Scores May Not Identify Good Models of the Brain

**Rylan Schaeffer**
Computer Science
Stanford University
rschaef@cs.stanford.edu

**Mikail Khona**
Physics
MIT
mikail@mit.edu

**Sarthak Chandra**
Brain & Cognitive Sciences
MIT
sarthakc@mit.edu

**Mitchell Ostrow**
Brain & Cognitive Sciences
MIT
ostrow@mit.edu

**Brando Miranda**
Computer Science
Stanford University
brando9@cs.stanford.edu

**Sanmi Koyejo**
Computer Science
Stanford University
sanmi@cs.stanford.edu

## Abstract

A prominent methodology in computational neuroscience posits that the brain can be understood by identifying which artificial neural network models most accurately predict biological neural activations, measured according to regression test error or other similar metrics. In this opinion piece, we argue that the field lacks a canonical definition of model goodness, and rather than engaging with this difficult question, the neural regressions methodology simply asserted a proxy – neural predictivity – then overfit to this proxy. We begin with a notable failure of the neural regressions methodology in which the most predictive models disagreed with key properties of the neural circuit. Next, we highlight converging empirical and mathematical evidence that explains the disconnect: (linear) neural regressions are simply discovering the implicit biases of (linear) regression, which may not appropriately identify models that are actually brain-like. This is an instance of Goodhart's law: by selecting neural network models that optimize (linear) neural predictivity, the field's results have devolved into re-discovering general properties of (linear) regression, rather than furthering our understanding of the brain. These insights suggest that the neural regressions methodology may be insufficient for understanding the brain, and we call for a critical reevaluation of this methodology in computational neuroscience.

## 1 Introduction

An influential methodology in neuroscience-inspired artificial intelligence argues that task-optimized deep artificial neural networks (ANNs) should be considered good models of the brain if they capture a large fraction of variance in neural population recordings assessed via regressions of ANN unit activity onto biological neural responses [89]. The claim is that the ANN(s) with better performing neural regressions are more similar to the brain than alternative models [71]. This approach has been widely used in vision [90, 24, 43, 50, 72, 92, 40, 88, 79, 64, 18, 45], audition [46, 82, 55, 80], language [62, 39, 73, 3, 61, 15, 16, 33, 5, 2, 38, 58, 17, 44, 4, 81, 56, 37], and spatial navigation [57], most often with (regularized) linear models, but occasionally with non-linear models.

In this position piece, we argue that Neuro-AI lacks sufficiently rich definitions of neural similarity, and such notions are context-dependent and difficult to quantify. The neural regressions methodology sidesteps these challenges by defining a proxy – for instance, the test $R^2$ of linear regression between biological recordings and model activations – and then choosing models based on this proxy. The

Published at Unifying Representations in Neural Models (UniReps) Workshop at NeurIPS 2024.

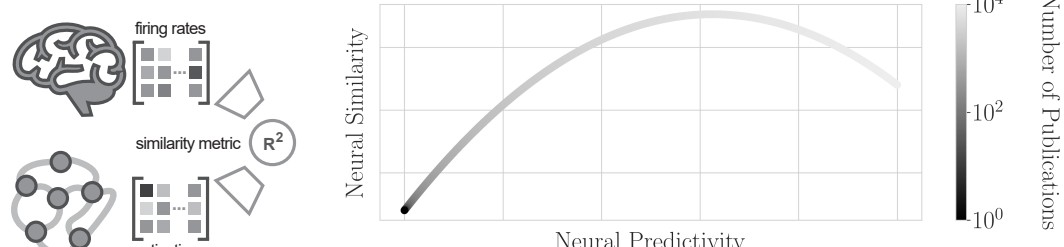

Figure 1: **Schematic.** Left: The neural regressions methodology posits a proxy – neural predictivity – of how similar model(s) are to a neural system of interest without verifying the extent to which the proxy agrees with neural similarity. Right: Overfitting to the proxy leads to mismatches with neural similarity. Although we do not define neural similarity here, we emphasize that it is task-, neural-system, and question-dependent, and hence likely cannot always be neural predictivity. For a system-specific example of neural similarity, we offer criteria for grid cells in Appendix Sec. A.

models that win a selection process (e.g., on BrainScore [71]) may do so more because of inductive biases of the proxy, independent of any meaningful relationships with the brain (Fig. 1).

This perspective explains why, for example, the neural regressions methodology was confidently incorrect when applied to models of grid cells: linear regression does not capture in key criteria of neural similarity for grid cells (e.g., periodic tuning curves [34], multiple grid modules with specific period ratios [77], toroidal continuous attractor dynamics [91, 32]; see Appendix Sec. A for a detailed list). This perspective also explains a finding by four independent research groups in different modalities, data, architectures and recording technologies [66, 25, 80, 17] of a quantitatively consistent relationship between test $R^2$ and effective dimensionality, that was mathematically corrected and further empirically studied by Canatar et al. [14]: (linear) neural predictivity *is* (linear) regression, and (linear) regression has inductive biases, irrespective of the underlying neuroscience. We focus on linear regression because of its ubiquity in the literature, but other preference functions (e.g., RSA [49], CKA [48], SVCCA [63], Procrustes [84], etc.) would not escape this critique; rather, they would simply change the inductive biases of the chosen preference function.

Together, these insights suggest that the neural regression methodology, and more broadly the idea that a uniform set of metrics can automate model selection, may be fundamentally flawed by overfitting to those metrics rather than advancing our understanding of the brain. We conclude by suggesting a re-evaluation of such methodologies.

## 2 Neural Regressions Can Reach Incorrect Conclusions with High Confidence

In vision, Bowers et al. [9] documented how artificial networks preferred by the neural regressions methodology lack or contradict properties of primate vision, and others have identified additional flaws [54, 88, 20, 35, 26, 27, 23, 36, 52]. Here, we chose to focus on the clearest example of a failure of the neural regressions methodology: grid cells. Why focus on grid cells? Grid cells – a surprising and important Nobel Prize-winning discovery [34] – differ from vision, audition and language in that humanity possesses scientific models [29, 11, 10, 76] that have repeatedly proven predictive ([77, 91, 32]), not in the regressions sense but in the sense of exhibiting fundamental properties, e.g., localization of each module to a two-dimensional subspace, quantization of grid module periods, preserved low-dimensional dynamics across waking and sleep that were subsequently validated. In a domain we understand well, how did the regressions methodology fare?

*When applied to a specific neural circuit (grid cells) that humanity possesses*
*near-normative models of, the neural regressions methodology preferred incorrect*
*models with high confidence.*

As context, the key research questions about grid cells are modeling their dynamics and the evolutionary causes for their existence. Previous and now near-normative models showed how strong recurrent interactions leading to pattern formation, coupled with a way for movement inputs to shift the pattern phase and thus perform path integration, could generate grid cell dynamics [11, 47]; and

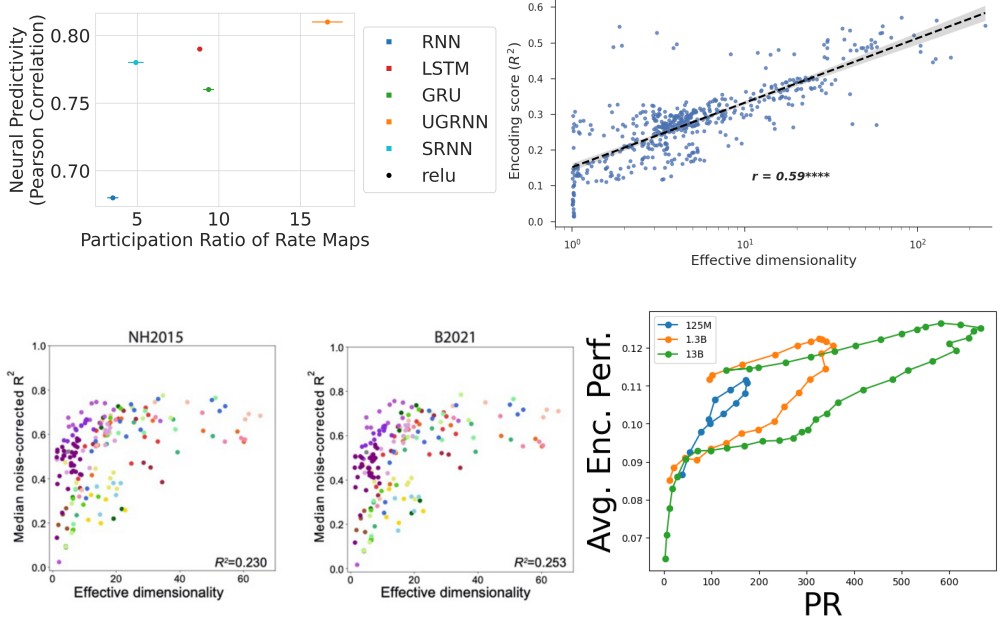

Figure 2: Four independent publications studying four different modalities and brain circuits in three different species found a consistent quantitative heuristic: Test $R^2$ is an affine transformation of the log participation ratio (Eqn. 2). Figures from Spatial Navigation in Mouse Medial Entorhinal Cortex [66], Vision in Macaque IT Cortex [25], Audition in Human Cortex. [80], Language in Human Cortex [17]. Later work [14] provided a spectral theory of the neural regressions methodology, which reveals results like these are attributable to *general properties of linear regression*, not the brain.

that multiple grid modules played key roles in disambiguating position over large ranges and in error correction [29, 76]. Later, deep recurrent networks trained in a supervised manner to path integrate were shown to learn grid-like units [7, 19, 74], and neural-regressions based work [57] showed that these supervised deep path integrators achieved the best performance possible at predicting recordings from mouse medial entorhinal cortex, leading the authors to call for better neural data.

However, multiple independent lines of evidence demonstrated that these high $R^2$ deep learning models are worse models of grid cells: (1) The required supervised targets, putative place cells, contradict known biological properties of place cells at both the single cell and population levels [67]; (2) The grid-like units lack key properties of real grid cells: periodic triangular tuning curves, multiple discrete grid modules, and specific ratios between grid modules [66, 68]; (3) the artificial grid units in some works were statistically indistinguishable from low pass-filtered noise [74, 75]. (4) In terms of evolutionary origins, the path integration objective of high-$R^2$ networks is not a sufficient objective for grid cells, as demonstrated in empirical deep neural network work [42, 41, 68], argued by prior theoretical work [28, 76, 53, 83], and shown by newer deep learning models [31, 85, 22, 70, 86, 87].

To summarize, the neural regressions methodology strongly supported deep learning-based path integrators because the networks achieved high neural predictivity scores, despite their discrepancies with multiple key criteria of neural similarity (listed in Appendix Sec. A). Why?

## 3 The Neural Regressions Methodology Reveals the Implicit Biases of Regression, Not Which Candidate Networks Are Similar to the Brain

Schaeffer et al. [66] made a conjecture: "different [models] achieve different neural predictivity scores because they learn different intrinsic dimensionalities, that then provide richer/poorer bases for linear regressions." Larger models simply provide more basis features for regression, and thus can provide better predictions independent of the similarity with the brain. To test their conjecture, the authors trained the same networks studied by Nayebi et al. [57] and empirically discovered that

reported test Pearson correlations exhibit an approximately linear-log relationship with a widely-used measure of effective dimensionality called participation ratio (PR) [21] (Fig 2a).

More precisely, consider $P$ stimuli, and denote artificial activations with $M$ units as $\mathbf{X} \in \mathbb{R}^{P \times M}$ and biological responses with $N$ neurons as $\mathbf{Y} \in \mathbb{R}^{P \times N}$. We fit linear models using $p < P$ data:

$$\hat{\beta}(p) \stackrel{\text{def}}{=} \underset{\beta \in \mathbb{R}^{M \times N}}{\arg\min} ||\mathbf{X}_{1:p}\,\beta - \mathbf{Y}_{1:p}||_F^2 + \alpha_{\text{reg}}||\beta||_F^2 \tag{1}$$

Letting $\mathbf{X}\mathbf{X}^T = \sum_{i=1}^{P} \lambda_i \mathbf{v}_i \mathbf{v}_i^T$, Schaeffer et al. [66] empirically found that approximately:

$$\text{Test } R^2 \sim \alpha \log(\text{Participation Ratio}) + \beta \quad ; \quad \text{Participation Ratio} \stackrel{\text{def}}{=} \frac{(\sum_{i=1}^{P} \lambda_i)^2}{\sum_{i=1}^{P} \lambda_i^2} \tag{2}$$

Participation ratio (PR) is a linear geometric measure of effective dimensionality: for uniform eigenvalues, the PR is the ambient dimensionality, whereas for a single non-zero eigenvalue, the PR is 1. Concurrent and subsequent work found quantitatively similar results across species, modalities, brain circuits and recording technologies: Elmoznino and Bonner [25] in deep convolutional networks trained on vision tasks to predict macaque IT cortex (Fig 2b), Tuckute et al. [80] in deep auditory networks to predict human brain-wide fMRI responses (Fig 2c), and Cheng and Antonello [17] in language models to predict human brain-wide fMRI responses (Fig. 2d). This finding by four independent research groups across different data modalities, tasks, architectures, species and recording technologies is puzzling. Are these results indicative of some deeper scientific insight into the brain?

In our view, no. *This pattern is attributable to the neural regressions methodology, not the brain.* Participation ratio (PR) was a reasonable first guess but an imprecise one that was subsequently refined into a more complete spectral theory of the regressions methodology. Canatar et al. [14] showed the normalized error $E_g(p)$ of *any* linear model $\hat{\mathbf{Y}}(p) \stackrel{\text{def}}{=} \mathbf{X}\hat{\beta}(p)$ is given as:

$$E_g(p) \stackrel{\text{def}}{=} \frac{||\hat{\mathbf{Y}}(p) - \mathbf{Y}||_F^2}{||\mathbf{Y}||_F^2} = \sum_{i=1}^{P} \frac{||\mathbf{Y}^T \mathbf{v}_i||_2^2}{||\mathbf{Y}||_F^2} \cdot \frac{\kappa^2}{1-\gamma} \frac{1}{(p\lambda_i + \kappa)^2}, \tag{3}$$

where $\gamma = \sum_{i=1}^{P} \frac{p\lambda_i^2}{(p\lambda_i + \kappa)^2}$ and $\kappa = \alpha_{\text{reg}} + \kappa \sum_{i=1}^{P} \frac{\lambda_i}{p\lambda_i + \kappa}$ must be solved self-consistently. This result says that the focus on PR by previous work was incomplete: PR partially captures the dimensionality of the learnable subspace, but the error *also* depends on the terms $\{||Y^T v_i||_2^2/||Y||_F^2\}_i$, which express whether the target $Y$ lies in that subspace. Thus, higher PR can be beneficial to express the task fully, but can also be harmful by being too expressive and then harming sample complexity. This partially explains why ZCA whitening to maximize participation ratio did not achieve exceptional neural predictivity, why increasing the number of covariates does not necessarily increase neural predictivity [25], and how randomly initialized networks can achieve high neural predictivity [45, 2].

While it may be tempting to think that these empirical results and this spectral theory of neural predictivity have taught us about the brain, note that this theory makes no assumptions about a neural, behavioral, biological, ethological or otherwise meaningful relationship between $\mathbf{X}$ and $\mathbf{Y}$. Rather, as its origin makes clear [8, 13, 12], this theory is fundamentally *a description of linear regression* [69]. Consequently, this leads to the following realization:

> *Taken to its extreme, the neural regressions methodology has taught us the implicit biases of our chosen proxy function (e.g., test $R^2$ of linear regression), not which candidate artificial neural networks are actually similar to the brain.*

## 4 Discussion

To summarize, NeuroAI lacks canonical definitions of neural similarity, and such notions are likely task-, system-, and question-dependent, as well as difficult to quantify. Rather than facing these challenges, the neural regressions methodology sidesteps them by defining a proxy – for instance, the test $R^2$ of linear regressions fit between biological recordings and artificial activations – and then choosing networks based on this proxy. The networks that win a selection process do so because of the proxy's implicit biases, independent of any meaningful relationship with the brain.

To explain with an analogy, in the field of language modeling, researchers want language models to generate responses preferred by humans. However, collecting human preferences is slow, costly and noisy, so researchers instead train modified language models called reward models to serve as proxies of human preferences. These reward models are a proxy for what we actually care about – human preferences – but the field is willing to use these proxies because the reward models are directly trained to emulate human preferences and are correlated with human preferences empirically [93, 78, 6, 60, 51]; even so, overfitting to the reward models at the expense of human preferences is still a commonly encountered problem [78, 30, 1].

In comparison, in computational neuroscience, researchers want models that are most similar to brain system(s) of interest. However, interacting with neural systems and running experiments is slow, costly and noisy, so researchers instead fit neural regressions to serve as proxies of neural similarity. These regressions are a proxy for what we actually care about – neural similarity – but in contrast with reward models, neural regressions are not trained to emulate neural similarity and have no known relationship with neural similarity.

To reiterate an earlier point, we focused on linear regression because of its ubiquity in the literature, but other proxies of neural similarity (e.g., RSA [49], CKA [48], SVCCA [63], Procrustes [84]) would not escape this critique; rather, *other proxies would simply change the pertinent implicit biases*.

Altogether, these insights suggest that the neural regressions methodology may be flawed, teaching us about the preferences that we as researchers implicitly chose instead of advancing humanity's understanding of the brain. We conclude by calling for a critical and careful re-evaluation in computational neuroscience of how the neural regressions methodology is used and interpreted. For a Future Outlook, please see Appendix Sec. B.

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

# A  Example Criteria of Neural Similarity to Grid Cells

In this paper, we intentionally do not provide a general definition of "neural similarity" (see Future Outlook - Appendix Sec. B), in part because we feel such a definition is likely highly context dependent. But we can offer a constructive example in the narrow context of grid and place cells viewed at the level of their circuit dynamics. When considering models, researchers often consider the following (non-exhaustive) list of relevant criteria for evaluating whether a model is similar to the circuit:

- Individual neurons exhibit equilateral triangular periodic tuning curves
- In the population of grid cells, multiple grid periodicities exist
- The periodicities of the grid cells are quantized
- The quantized periods of the modules exhibit precise ratios between adjacent periods
- The population states of each grid module (subpopulation with common period) lie on the surface of a 2D torus.
- The cell-cell relationships of co-modular grid cells (and their toroidal population states) are invariant across spatial environments and behavioral states.
- In any environment, grid-like tuning is present from the first trajectory fragment.
- While grid cells remain invariant in their relationships across environments, place cells remap or scramble their relationships.

# B  Future Outlook

Despite our critiques of the neural regressions methodology, model-system comparison is a fundamental and necessary component of a modeling science. How, then, can we move beyond flaws arising as a consequence of emphasizing only a single metric?

One possibility is to use a number of different comparisons that emphasize different aspects of model and system. This may include comparing behavior on top of neural activations, as is already a feature of the Brain-Score platform ([89, 71]), neural dynamics on top of neural geometry [59], or using a variety of different metrics that have different biases ([35]). Beyond linear regression, computational neuroscience has introduced a number of other candidates into the literature, including RSA [49], Procrustes [84], CKA [48], SVCCA [63], and a number of variants of these metrics. All of the above metrics compare geometric features of neural activations. Recently proposed methods, such as Dynamical Similarity Analysis (DSA, [59]) compare different features of the system, like dynamical structure. Older work sought to study similarity using combined perspectives of behavior, representations, dynamics and circuit mechanisms, e.g., [65]. Using more types of comparison, both in terms of metrics and data, should help mitigate the biases of individual comparisons, making Goodharting more challenging. However, it is important to note that even combinations of such metrics are liable to fall prey to Goodhart's law. Depending on the scientific question, the relevant quantity to be compared may change.

More generally, beyond significantly increasing the number of types of comparisons being done, it is worth taking a step back and asking what we mean by a 'good model'. Although we do not define neural similarity here, in our view, neural similarity depends on the task, the neural system and the particular scientific question. Thus, in our opinion, neural similarity cannot always be neural predictivity. This perspective lays bare the difficulty and, in a sense incoherence, of seeking to globally define good models in terms of one or even a set of metrics.

