# OpenReview forum: "Position: Maximizing Neural Regression Scores May Not Identify Good Models of the Brain"
_NeurIPS.cc/2024/Workshop/UniReps — UniReps_

### Official Review · Reviewer_6g4W · 2024-10-02

**Rating:** 7
**Confidence:** 3

**Review:**

This work calls into question the framework of evaluating models of the brain based on their ability to predict neural activity in the brain using linear regression on the model representations. The authors first use grid cells as evidence that the regression framework fails. They argue that models that perform well at predicting grid cell response (in terms of linear regression) lack the well-studied characteristics of grid cells. The authors then argue that the goodness of fit of linear regressions are more related to implicit biases of linear regression rather than a model capturing the relevant properties of neural responses. Specifically, they cite theoretical work demonstrating that the participation ratio (PR) of the model responses is an important factor in determining the performance a linear regression task, as well as empirical work showing that performance of linear regression correlates with PR.

Strengths: Overall I enjoyed reading this paper. It raises important consideration and questions about using a linear regression framework to evaluate models of the brain.

Relevance: This work is a clear fit for this workshop, therefore I recommend that it be accepted.

Question: Based on the authors' second argument that much of the performance of linear regression can be explained by the PR of model responses, why not just control for the PR when using linear regression as a measure of performance? Given the theory supporting the authors' argument, it seems like there should be a principled approach to this.

Minor: there is a missing equation reference in Figure 2's caption.

---

### Official Review · Reviewer_ZVey · 2024-10-04
**An interesting and provocative paper that needs more detail**

**Rating:** 8
**Confidence:** 3

**Review:**

This is an interesting brief report arguing that the high alignment in neural predictivity between higher dimensionality models is due to linear regression rather than properties of the networks themselves. I think this paper makes an interesting point, but while some aspects were clear others were harder to follow and felt like they need more justification. For example, it is not clear that methods of neural alignment do not account for similarity in eigenvectors/values and regression targets. Further, most of the references cited arguing for higher effective dimensionality show that simply increasing the number of regressors does not increase linear predictivity, and I had a hard time reconciling these results with this paper. Further, while there is a brief discussion of other methods in the appendix, it was not entirely clear whether the arguments extended to other common types of neural similarity / mapping measures.

Overall I think this is an interesting paper that will be a good addition to the workshop.

---

> ### Author Response · Authors · 2024-10-19
> **Response to Reviewer ZVey's Review of Submission 31**
>
> UniReps has graciously given submitting authors the opportunity to respond to reviewers, so we wanted to thank you for reviewing our paper. We submitted the work to workshops to begin a conversation and to solicit feedback, and we want to highlight how we've incorporated your feedback into our work.
>
> > Further, most of the references cited arguing for higher effective dimensionality show that simply increasing the number of regressors does not increase linear predictivity, and I had a hard time reconciling these results with this paper.
>
> We rewrote a subset of Section 3 to clarify that we are not arguing that higher effective dimensionality monotonically increases neural predictivity. The two points we wished to communicate in Section 3 is (1) that the community discovered an interesting but *imprecise* phenomenon that was later theoretically and empirically refined to clarify that higher ED can help _or hurt_ in specific contexts, but that (2) these ED discoveries and ED theory are properties of _linear regression_, independent of neuroscience.
>
> If you can suggest how we might better express this perspective, we would be keen to hear your feedback!

---

### Official Review · Reviewer_65GB · 2024-10-06
**Neural predictivity does not imply similarity**

**Rating:** 4
**Confidence:** 5

**Review:**

This opinion piece claims a common misconception in NeuroAI is that better neural predictivity (from Artificial Neural Networks (ANNs) representations) implies higher representational similarity to the brain (measured in terms of linear predictivity, RSA, CKA, shape metrics, etc.). Taking the example of grid cells, the authors cite the ANN models that do not exhibit grid cell-like properties but still achieve good neural predictivity. Furthermore, the authors cite converging evidence from various fields that demonstrate a relationship between the neural predictivity ($R^{2}$) and intrinsic dimensionality of ANNs (measured in terms of participation ratio). Overall, the authors call for a critical re-evaluation of the neural regressions methodology.

I agree with the author's opinion that predictivity does not imply similarity, however, I believe clearer and more explicit evidence would improve the paper towards removing this misconception. Some of the things that can be improved are as follows:
  1. The authors state that the definition of neural similarity and model goodness is context-dependent (line 30). At the end of section 2, the authors claim that highly predictive models are worse models of grid cells, but fail to provide the context of the implied model goodness. Intuitively, these ANN models are worse in terms of implementing properties of grid cells but are better in terms of predicting their activity. Rather than ending this section with a question, explicitly stating that neural regressions methodology supported ANN models because of predictivity rather than similarity, would provide a better takeaway for the reader.
  2. The sub-heading of section 3 can be improved. The authors state at the start of the paper that the neural regressions methodology is about whether and which ANNs are more similar to the brain. This implies that any insights gained from this methodology would be about the models and not about the brain. Hence, it is intuitive that the methodology does not reveal insights into the brain, but is rather believed to provide insights about model goodness. I recommend rephrasing the heading (similar to the last line in section 3) to be more explicit about what the methodology can and can not do.
  3. In section 3, the authors provide evidence that neural predictivity is dependent on the intrinsic dimensionality of ANNs. However, the link to neural similarity here is not explicitly stated. The effect of intrinsic dimensionality on neural similarity is not intuitive, but I believe it is implied that they are independent. Explicitly showing the effect of effective dimensionality on neural predictivity and neural similarity would greatly improve this section by providing better evidence for the claim.
  4. Similar to point 3, I believe the title of the paper can also be improved to explicitly state that maximizing neural regression scores does not teach us about which model is better/model goodness (or a similar statement framed as a question).
  5. At the end of the discussion, the authors state that the neural regressions may be fundamentally flawed. I believe that the methodology is not flawed, but the takeaways that some researchers take from it are flawed. It might be useful to highlight the context in which the methodology is flawed (in terms of similarity), or better state the cases where researchers may use this methodology (neural response prediction) and the cases where this should be avoided (estimating similarity).

I believe these fixes are very important to provide a clear message to the readers and avoid further confusion in the line of research which is already riddled with misconceptions (eg., the one stated in the paper). In the absence of a rebuttal period to improve the paper, I, unfortunately, opt for a lower rating. However, when improved, I believe this can be a crucial and important paper in guiding the field.

Other questions/suggestions:
  1. It is unclear to me how Figure 1 Right was obtained. If it is a schematic figure, explicitly state it.
  2. Figure 2: Label subplots (a-e)
  3. Figure 2 caption, typo: Eqn ??
  4. Line 94, mention that PR is the participation ratio, as PR is used several times without explicitly defining the acronym
  5. Line 421, fix RSA citation
  6. The first paragraph of the discussion is very similar to the second paragraph in the introduction. I recommend rephrasing for better use of limited space.

---

> ### Author Response · Authors · 2024-10-19
> **Response to Reviewer 65GB's Review of Submission 31**
>
> UniReps has graciously given submitting authors the opportunity to respond to reviewers, so we wanted to thank you for reviewing our paper. We submitted the work to workshops to begin a conversation and to solicit feedback, and we want to highlight how we've incorporated your feedback into our work.
>
> - "Rather than ending this section with a question, explicitly stating that neural regressions methodology supported ANN models because of predictivity rather than similarity, would provide a better takeaway for the reader." -> We rewrote the end of Section 2 to explicitly state this divergence.
>
> - "I recommend rephrasing the heading (similar to the last line in section 3) to be more explicit about what the methodology can and can not do." -> We changed the text of Section 3 to adhere more closely to the ending of the section
>
> - "I believe the title of the paper can also be improved to explicitly state that maximizing neural regression scores does not teach us about which model is better/model goodness (or a similar statement framed as a question)." We changed the title of the manuscript and also preappended "Position" to clarify that this manuscript is a position piece.
>
> - "At the end of the discussion, the authors state that the neural regressions may be fundamentally flawed. I believe that the methodology is not flawed, but the takeaways that some researchers take from it are flawed." We think this is a valuable nuance that is much appreciated. We will work to integrate this into the manuscript as we work towards a full conference submission.
>
> - "It is unclear to me how Figure 1 Right was obtained. If it is a schematic figure, explicitly state it." -> We explicitly labeled Figure 1 as a *Schematic*
>
> - "Figure 2 caption, typo: Eqn ??" We fixed the incorrect equation reference
>
> - "Line 94, mention that PR is the participation ratio, as PR is used several times without explicitly defining the acronym" -> Done
>
> - "Line 421, fix RSA citation" -> Done. Thank you for your keen eye.

---

> > ### Comment · Reviewer_65GB · 2024-10-23
> >
> > I appreciate the author's response and believe the paper's quality has improved. It has definitely started a conversation in the field that would be very helpful to future researchers.

---

### Official Review · Reviewer_bHu3 · 2024-10-07
**Review of Submission 31**

**Rating:** 6
**Confidence:** 4

**Review:**

**Quality:** While I do tend to agree with the overall assessment that neural regression scores alone do not necessarily indicate greater similarity to the brain (and that "similarity" to the brain is itself context-dependent and not well-defined), I think a good chunk of the criticism here comes from the debate surrounding theories of the emergence of grid cells -- a debate that I don't think is definitively resolved in favour of one side (there are several responses to the Schaeffer et al. works from Sorscher et al. with defences of the latter's theory).

Section 3 seems to provide more compelling arguments in my opinion, but it is worth noting that papers such as Elmoznino & Bonner [20 in the paper] do not claim that DNN models with high-dimensional representations are better models of the visual cortex, they explicitly state that they are just able to predict neural activity better. From that paper:

> ... we found that the relationship between latent dimensionality and encoding performance generalized across layer depth, meaning that even within a single layer of a DNN hierarchy, encoding performance can widely vary as a function of latent dimensionality.

> ... while our findings show that computational models of visual cortex benefit from high latent dimensionality, our method cannot speak to the dimensionality of visual cortex itself and was not designed to do so ... high-dimensional DNNs should generally better explain neural activity even if neural representations in the brain are low-dimensional.

which do not seem like claims that models with high-dimensional representations are more similar to visual cortex than others. In that sense, it seems to align well with the argument made in Section 3 that higher dimensionality should improve neural predictability.

Finally, the paper calls for greater nuance in defining neural similarity in a context-dependent manner, which is appreciated. Indeed, one may find great similarity in the behaviour of two systems but the neural dynamics underlying these could be entirely different -- so comparisons must be made at multiple levels to be comprehensive. Suppl. Section B discusses some metrics that could provide for more comprehensive comparisons than linear regressions (while acknowledging the potential for Goodharting in those cases as well), but some work that explicitly discusses the flaws in these metrics could be cited, e.g., Davari et al., Dujmovic et al., Helmer et al. among others.

Davari, M., Horoi, S., Natik, A., Lajoie, G., Wolf, G., & Belilovsky, E. Reliability of CKA as a Similarity Measure in Deep Learning. In The Eleventh International Conference on Learning Representations.

Dujmovic, M., Bowers, J., Adolfi, F., & Malhotra, G. Inferring DNN-Brain Alignment using Representational Similarity Analyses can be Problematic. In ICLR 2024 Workshop on Representational Alignment.

Helmer, M., Warrington, S., Mohammadi-Nejad, A. R., Ji, J. L., Howell, A., Rosand, B., ... & Murray, J. D. (2024). On the stability of canonical correlation analysis and partial least squares with application to brain-behavior associations. Communications Biology, 7(1), 217.

**Clarity:** The writing is clear, however, I did spot a few issues:
1. Figure 1 is not referenced in the text, and the caption for the right panel does not clearly indicate whether it is just a schematic or an actual plot. When the paper argues that "neural similarity" isn't well-defined, it's unclear to me what the neural similarity on the y-axis is. It's also unclear what "number of papers" is referring to here.
2. Figure 2 caption: Missing reference to equation.
3. Line 421: Missing reference/in-text citation.
4. Relatively minor, Goodhart's law finds mention in the abstract and appendix but never in the main text.

**Originality:** I'm not sure whether discussing the originality of this submission matters, or how best one would judge originality in the context of a perspective paper. Criticism of regression-based neural similarity metrics has existed for a while and several works have addressed this, but this work attempts to consolidate the evidence and provide a coherent viewpoint: that "neural similarity" is not well-defined.

**Significance:** This submission would definitely elicit discussion at the workshop on what "neural similarity" is and how best to judge this in different contexts. It could also motivate researchers to be cautious and comprehensive when claiming similarity of models to brains.

---

> ### Author Response · Authors · 2024-10-19
> **Response to Reviewer bHu3's Review of Submission 31**
>
> UniReps has graciously given submitting authors the opportunity to respond to reviewers, so we wanted to thank you for reviewing our paper. We submitted the work to workshops to begin a conversation and to solicit feedback, and we want to highlight how we've incorporated your feedback into our work.
>
> - We included citations for the relevant prior work we missed that you recommended (Davari, Dujmovic, Helmer)
> - We added a reference to Figure 1 in the text, and we clearly labeled Figure 1 as a **Schematic**
> - We fixed the missing reference to an equation in Figure 2's caption
> - Regarding "Relatively minor, Goodhart's law finds mention in the abstract and appendix but never in the main text.", this is a good point that we agree with. The feedback will help us shape this workshop submission into a more full submission.
>
> > I think a good chunk of the criticism here comes from the debate surrounding theories of the emergence of grid cells -- a debate that I don't think is definitively resolved in favour of one side (there are several responses to the Schaeffer et al. works from Sorscher et al. with defences of the latter's theory)
>
> *If we can solicit your feedback*, would it be helpful if we were to walk through the history of this debate and why we believe the evidence resolves the debate in favour of one side? We felt like recapitulating that history and the evidence might distract from the focus on neural regressions, but perhaps readers might appreciate a comprehensive appendix with a deep dive into the grid cell debate.
>
> > a debate that I don't think is definitively resolved in favour of one side (there are several responses to the Schaeffer et al. works from Sorscher et al. with defences of the latter's theory).
>
> If you have time and interest, you might find the latest response informative: https://arxiv.org/abs/2312.03954

---

### Public Comment · ~Aran_Nayebi2 · 2024-10-17
**Many Unsubstantiated Claims**

This paper contains several misconceptions that are worth mentioning, which I’ve also shared on Twitter: https://x.com/aran_nayebi/status/1846689522932761029. It seems these points were missed by the reviewers (though the oversight responsibility remains solely with the authors), which is why I’m also posting it on here.

**In short, unlike what the authors claim, effective dimensionality is *not* the whole story for model-brain linear regression, for several reasons:**

(1) The spectral theory due to [Canatar*, Feather* *et al.* 2023](https://arxiv.org/abs/2309.12821) that they cite is misrepresented in this paper as not making any assumptions about a meaningful relationship between the model ($X$) and the neural data ($Y$). However, this theory crucially relies on model alignment with brain data. This is already evident from their equations, as it quite literally includes an alignment term ($Y^Tv_i$) of the model's principal components $v_i$ being aligned with the neural data ($Y$). In fact, [Canatar*, Feather* *et al.* 2023](https://arxiv.org/abs/2309.12821) conclude that effective dimensionality alone doesn’t fully predict neural data (cf. their Figures SI5.10-SI5.12).

(2) Even in this paper’s ***own*** Figure 2 (top left panel), networks with **low participation ratios**, like [Nayebi *et al.* 2021](https://www.biorxiv.org/content/10.1101/2021.10.30.466617v2)'s SRNN, still achieve high neural predictivity of medial entorhinal cortex (MEC), already indicating that participation ratio is not a unilateral predictor of whether a model will match the brain via linear regression.

(3) Beyond MEC, this lack of a trend with effective dim is also the case when looking at models in their match to macaque IT, human OTC, and human auditory cortex. For example, [Conwell *et al.* 2023](https://biorxiv.org/content/10.1101/2022.03.28.485868v2)’s very thorough work involving 1.8 billion model-brain regressions shows that effective dim isn’t related to linear prediction of human OTC (cf. their Figure 5 w/ linearly weighted RSA). Moreover, in macaque IT, as Ratan Murty & DiCarlo have seen, if you include very predictive models of IT responses, the effective dim to linear predictivity trend also doesn’t hold (this observation for vision models is also corroborated in Figures SI5.10-SI5.12 of [Canatar*, Feather* *et al.* 2023](https://arxiv.org/abs/2309.12821)). Finally, in auditory cortex, specifically the work of [Tuckute*, Feather* *et al.* 2023](https://journals.plos.org/plosbiology/article?id=10.1371/journal.pbio.3002366) that this paper cites, their Figure S9 shows that effective dim does *not* explain all the variance among models either.

(4) Furthermore, the above models that this paper cites participation ratios for, were also compared by their original authors against several ***non-fitted*** metrics (like RSA, score distributions, simpler-than-linear mappings, etc.), which found similar conclusions that matched linear regression results, across MEC (cf. Figure 3 of [Nayebi *et al.* 2021](https://www.biorxiv.org/content/10.1101/2021.10.30.466617v2)), IT (cf. [Khaligh-Razavi & Kriegeskorte 2014](https://journals.plos.org/ploscompbiol/article?id=10.1371/journal.pcbi.1003915) as the first of many such demonstrations), and auditory cortex (cf. Figure S10 of [Tuckute*, Feather* *et al.* 2023](https://journals.plos.org/plosbiology/article?id=10.1371/journal.pbio.3002366)).

(5) While one often discusses trained models, the belief that a randomly initialized neural network *shouldn't* match the brain well is a common misconception. This is because a randomly initialized neural network is a highly *non-random* function. After all, the architectures one invents are often designed so that they will do well when optimized on a task.

(6) Although the evidence in this paper doesn’t support the strong claim that "NeuroAI is overfitting to linear regression", it’s natural to ask what constitutes "good" in neuroscience. Regardless of preferences, it’s *objectively* better to predict the brain than not at all. Even if predictivity isn’t one's end goal, it’s still important to perform subsequent analyses on a predictive model than a non-predictive one, like in mechanistic interpretability analyses (e.g., our [Neuron 2023](https://pubmed.ncbi.nlm.nih.gov/37451264/) work). In the era of task-performant ANNs, we often overlook that prediction has long been a high bar in brain science, where many theories that prioritized mathematical interpretability failed to predict large-scale neural activity (e.g. Gabors & macaque V1, HMAX & macaque IT, grid-cell-only attractors & rodent MEC). Going forward, we think the aim should be to pick sharp transform classes where animals are aligned, as we want the models to be as predictive as any animal is to another. See our forthcoming work on this [here](https://x.com/cogphilosopher/status/1821551011393077480).

---

> ### Author Response · Authors · 2024-10-20
> **Responding to "Many Unsubstantiated Claims" by Dr. Nayebi**
>
> While we appreciate the interest, these comments misunderstand or misrepresent both our work and prior work.
>
> > In short, unlike what the authors claim, effective dimensionality is not the whole story for model-brain linear regression, for several reasons:
>
> Contrary to the implication, the manuscript actually state that effective dimensionality is not the whole story: “[Effective dimensionality] was a reasonable first guess but an imprecise one that was subsequently refined into a more complete spectral theory of the regressions methodology.” and “Thus, higher [ED] can be beneficial to express the task fully, but can also be harmful by being too expressive and then harming sample complexity. This partially explains why ZCA whitening to maximize participation ratio did not achieve exceptional neural predictivity, why increasing the number of covariates does not necessarily increase neural predictivity [25]”
>
> Section 3 should not be understood as endorsing conjectures by early work in this area; rather, we are offering a historical retrospective of the field to substantiate our claim that “the field’s results have devolved into re-discovering general properties of (linear) regression.”
>
> > The spectral theory due to Canatar*, Feather* et al. 2023 that they cite is misrepresented in this paper as not making any assumptions about a meaningful relationship between the model ($X$) and the neural data ($Y$). However, this theory crucially relies on model alignment with brain data. This is already evident from their equations, as it quite literally includes an alignment term
>
> This response either misunderstands the spectral theory or our manuscript. The task-alignment is not a property of the brain or candidate artificial networks. Rather, the task-alignment is a geometric property of arbitrary covariates and targets that describes how well linear regression will generalize; consequently, the task-alignment describes generalization of linear regression, independent of the brain.
>
> That the spectral theory is a statement about linear regression is explicitly stated by the original authors in an [earlier paper](https://proceedings.neurips.cc/paper/2021/hash/691dcb1d65f31967a874d18383b9da75-Abstract.html): “Our result is general in that it applies to any kernel, data distribution and target function.” This is why we write, “this theory makes no assumptions about a neural, behavioral, biological, ethological or otherwise meaningful relationship between X and Y”.
>
> Dr. Nayebi's misinterpretation of the spectral theory has been (in our view) [confirmed by the senior author of the spectral theory](https://twitter.com/CPehlevan/status/1847325545115406472).
>
> To clarify, we are not criticizing inductive biases. Rather, we are pointing out that Goodharting has shifted the field from offering insights into the system of interest (the brain) to insights into the preference function (e.g., test R^2 of linear regression).
>
> > Furthermore, the above models that this paper cites participation ratios for, were also compared by their original authors against several non-fitted metrics (like RSA, score distributions, simpler-than-linear mappings, etc.), which found similar conclusions that matched linear regression results
>
> Our perspective critiques how neural comparisons are currently used and interpreted. Though the current opinion focuses on linear regression, we will extend a future version of this work to encompass neural comparisons generally (including RSA).
>
> To add nuance, using multiple neural comparisons (e.g., linear regression + RSA) is not inherently good and cannot be relied upon as a universal sanity check. Rather, multiple comparisons should be used to complement and correct for one another’s inadequacies. For example, in the grid cell literature, a common comparison between neurons and units is how grid-like the tuning curves are; however, this grid score is imperfect because it scores rotational symmetry - not hexagonal lattices - so researchers use multiple complementary methods (e.g., topological data analysis, circuit identification).
>
> **To briefly restate our overarching perspective, comparing a system and a model requires specifying what similarity to that system means. Once “neural similarity” has been defined, one can then identify proxy metrics for neural similarity (e.g., test R^2 of linear regression). However, as the field optimizes against a fixed proxy metric, the field begins Goodharting: overfitting to the proxy at the cost of true “neural similarity” (which we believe is a neural system-, setting- and task- dependent notion).**
>
> Solutions include: (1) clearly identifying what similarity to a specific neural system means, (2) choosing bespoke metrics for the neural system of interest rather than a fixed generic metric, (3) identifying specific limitations of each metric in that specific context + ensembling different metrics to correct for other metrics’ shortcomings in that specific context.

---

> ### Public Comment · ~Aran_Nayebi2 · 2024-10-21
> **Responding to the authors' reply**
>
> I appreciate the response, but this paper's claims that the field has "devolved into overfitting", along with the implication that highly linearly predictive models are able to achieve this "*independent* of the brain", are nonetheless inaccurate from both empirical & theoretical perspectives:
>
> (1a) While it's true that every metric has its biases, it's well-established that even *despite* the differences between regressed and non-fitted metrics, there is generally agreement between highly linearly predictive models and these metrics, across brain areas & species (see my previous post for the relevant figures).
>
> (1b) Moreover, as related to the above, what we see emerging from these models often mirrors known features of neural processing in visual, auditory, motor, entorhinal, and other brain areas—such as Gabors, hierarchical structure, tuning properties, sparsity, and functional responses. That's not to say the models are perfect (this is even evident from our metrics), but this is a reflection that the metrics aren't pointing us *away* from the brain.
>
> (2) From an oracle perspective, neural responses in one animal’s brain area overlap in their encoding with the responses in the same brain area of another animal. This consistency is mirrored in our linear prediction metrics, and correctly provide an *upper bound* on model-brain neural predictivity—this is the "inter-animal consistency" referenced in the papers I shared in my prior post.
>
> (3) Regarding the spectral theory, the post from Cengiz that you linked is explaining that this theory is not intended as a framework for model comparison. It does *not* suggest there is zero overlap between models and the brain when neural data is used as a target. In fact, Canatar*, Feather* et al. 2023's work, which builds on his, directly manipulates task-model alignment and spectral properties to determine whether strong predictive scores are driven by the spectrum or alignment. Their conclusions do not support the idea of zero model alignment with the brain, as mentioned above.
>
> (4) The "solutions" the authors offer are exactly what the field of NeuroAI has been doing since its inception, as clearly indicated by all the papers posted above.

---

### Decision · Program_Chairs · 2024-10-10

**Decision:**

Accept

**Comment:**

In light of the positive reviewers' feedback and relevancy of the submission, we are pleased to accept this paper for presentation at UniReps 2024. We kindly ask the authors to incorporate the reviewers' suggestions and feedback in the final camera-ready version of the manuscript.